# A Systematic Review of the Prevalence of Persistent Gastrointestinal Symptoms and Incidence of New Gastrointestinal Illness after Acute SARS-CoV-2 Infection

**DOI:** 10.3390/v15081625

**Published:** 2023-07-26

**Authors:** Michael J. Hawkings, Natasha Marcella Vaselli, Dimitrios Charalampopoulos, Liam Brierley, Alex J. Elliot, Iain Buchan, Daniel Hungerford

**Affiliations:** 1Department of Public Health, Policy & Systems, Institute of Population Health, University of Liverpool, Liverpool L69 3GF, UK; 2National Institute for Health and Care Research Health Protection Research Unit in Gastrointestinal Infections, University of Liverpool, Liverpool L69 7BE, UK; n.vaselli@liverpool.ac.uk (N.M.V.);; 3Department of Clinical Infection, Microbiology and Immunology, Institute of Infection, Veterinary & Ecological Sciences, University of Liverpool, Liverpool L69 7BE, UK; 4Department of Health Data Science, Institute of Population Health, University of Liverpool, Liverpool L69 3GF, UK; 5Real-Time Syndromic Surveillance Team, Field Services, Health Protection Operations, UK Health Security Agency, Birmingham B2 4BH, UK

**Keywords:** SARS-CoV-2, long-COVID, post-COVID syndrome, gastrointestinal, diarrhoea, irritable bowel syndrome

## Abstract

It is known that SARS-CoV-2 infection can result in gastrointestinal symptoms. For some, these symptoms may persist beyond acute infection, in what is known as ‘post-COVID syndrome’. We conducted a systematic review to examine the prevalence of persistent gastrointestinal symptoms and the incidence of new gastrointestinal illnesses following acute SARS-CoV-2 infection. We searched the scientific literature using MedLine, SCOPUS, Europe PubMed Central and medRxiv from December 2019 to July 2023. Two reviewers independently identified 45 eligible articles, which followed participants for various gastrointestinal outcomes after acute SARS-CoV-2 infection. The study quality was assessed using the Joanna Briggs Institute Critical Appraisal Tools. The weighted pooled prevalence for persistent gastrointestinal symptoms of any nature and duration was 10.8% compared with 4.9% in healthy controls. For seven studies at low risk of methodological bias, the symptom prevalence ranged from 0.2% to 24.1%, with a median follow-up time of 18 weeks. We also identified a higher risk for future illnesses such as irritable bowel syndrome, dyspepsia, hepatic and biliary disease, liver disease and autoimmune-mediated illnesses such as inflammatory bowel disease and coeliac disease in historically SARS-CoV-2-exposed individuals. Our review has shown that, from a limited pool of mostly low-quality studies, previous SARS-CoV-2 exposure may be associated with ongoing gastrointestinal symptoms and the development of functional gastrointestinal illness. Furthermore, we show the need for high-quality research to better understand the SARS-CoV-2 association with gastrointestinal illness, particularly as population exposure to enteric infections returns to pre-COVID-19-restriction levels.

## 1. Introduction

Since its emergence in late 2019, the COVID-19 pandemic has resulted in over 750 million infections and 6 million deaths worldwide as of July 2023 [1]. While most cases are experienced as a self-limiting respiratory tract infection, some individuals may develop a more severe or protracted illness. Post-COVID syndrome, or long-COVID, refers to symptoms that persist beyond 12 weeks from the onset of infection [2]. In the UK, self-reported health data indicate that 2.1 million people, or 3.3% of the general population, were suffering from post-COVID in January 2023 [3]. Symptoms of post-COVID syndrome appear to affect every body system, often overlapping and with a significant mental health impact [2].

It is established that SARS-CoV-2 (the virus that causes COVID-19 disease) infection can result in gastrointestinal symptoms: in 2020, a meta-analysis of 8302 patients identified diarrhoea in 12% of paediatric and 9% of adult cases [4]. Elshazli et al. also identified gastrointestinal symptoms in 20% of 25,252 patients in 2020, with anorexia, dysgeusia, diarrhoea, nausea and haematemesis being the most common [5]. In Liverpool, UK, gastrointestinal symptoms were observed in a third of hospitalised COVID-19 patients [6]. Research investigating gastrointestinal symptoms as a component of post-COVID syndrome, however, is less substantial. One review of post-COVID syndrome patients identified persistent symptoms of dysgeusia and diarrhoea at a frequency of 7% and 6%, respectively [7]. Similarly, an electronic health record study of 273,618 individuals in the USA found persistent gastrointestinal symptoms in 8.3% of participants six months after COVID-19 onset [8]. Chopra et al. also found that patients in India presenting with diarrhoea during acute COVID-19 were more likely to suffer from post-COVID syndrome symptoms such as fatigue, dyspnoea and chest discomfort [9].

When considering how SARS-CoV-2 interacts with the gastrointestinal system and how this may result in persistent post-viral symptoms, several pathophysiological mechanisms have been proposed. SARS-CoV-2 initially binds with the angiotensin-converting enzyme-2 (ACE-2) receptor, which is highly expressed by ileal enterocytes [10,11]. The binding of SARS-CoV-2 to ACE-2 receptors may disrupt angiotensin homeostasis and reduce tryptophan absorption, resulting in inflammation and alteration of the gut microbiota [12,13]. Alteration of the gut microbiome is known to occur during acute COVID-19 infection, as it does in many other viral respiratory tract infections [14,15]. This can result in the depletion of beneficial gut commensals and the proliferation of opportunistic pathogens in the gastrointestinal tract [16]. Zuo et al. found this dysbiotic state to worsen over time, even after recovery from the acute illness, in moderate to severe hospitalised COVID-19 cases [17]. Phetsouphanh et al. identified raised levels of IFN-β, IFN-λ1 and highly activated innate immune cells eight months after COVID-19 diagnosis, while other studies detected the persistence of SARS-CoV-2 viral matter in the intestinal epithelium beyond recovery [18,19,20]. Another mechanism that has been recently suggested is the formation of fibrinolysis-resistant amyloid microclots and platelet pathology in post-COVID syndrome patients [21]. Kell et al. proposed that these microclots may impair tissue perfusion and be a key determinant of post-COVID syndrome [22].

Although, to the best of the authors’ knowledge, no studies thus far have investigated the mental well-being impact of persistent gastrointestinal post-COVID symptoms, the psychosocial impact of functional gastrointestinal disorders (FGIDs) is known to be significant. The similarity between FGIDs and post-COVID gastrointestinal symptoms can be drawn in their clinical features and poorly understood causal mechanisms, both likely including the intestinal microbiota and gut–brain axis [23]. Around half of irritable bowel syndrome (IBS) patients are thought to have concurrent mental health conditions, and symptom-specific anxiety can impair functions of daily life in affected patients [24]. The development of low mood and anxiety states has also been observed in post-COVID syndrome [25].

Given the potential for a substantial physical and mental disease burden of both post-COVID syndrome and gastrointestinal illness and the lack of clinical guidelines on managing post-COVID gastrointestinal sequelae, we conducted a systematic review aiming to summarise the current evidence in two areas. Firstly, to provide an overview of the prevalence of persistent gastrointestinal symptoms following acute SARS-CoV-2 infection, and secondly, to estimate the incidence of newly diagnosed gastrointestinal illness following recent SARS-CoV-2 infection (not including gastrointestinal complications of acute COVID-19).

## 2. Methods

### 2.1. Search Strategy

The review was conducted in accordance with the Preferred Reporting Items for Systematic Reviews and Meta-Analyses (PRISMA) standards for systematic reviews and registered with the International Register of Systematic Reviews (PROSPERO reference CRD42022315792, available from: https://www.crd.york.ac.uk/prospero/display_record.php?ID=CRD42022315792, accessed on 23 July 2023).

We conducted a systematic search of the literature using OVID MedLine, SCOPUS and Europe PubMed Central from1 December 2019 to 3 July 2023. We also searched pre-publication and the other literature indexed in medRxiv. Search terms were constructed around three themes: COVID-19, gastrointestinal symptoms or illness and observational study designs. We did not include terms such as ‘obesity’ due to the volume of unsuitable studies returned. Our search terms were reviewed by all authors and a health science librarian. Our full search strategy is shown in Table 1. Exact searches used for each database are provided in the Appendix A.

### 2.2. Inclusion/Exclusion Criteria

Studies were included in the review if they followed up with participants for ongoing gastrointestinal symptoms or new gastrointestinal illnesses beyond acute COVID-19 infection. We used the following inclusion criteria: (1) follow-up of gastrointestinal symptoms and the development of new gastrointestinal illness from 4 weeks after COVID-19 diagnosis or onset, as per the National Institute for Health and Care Excellence (NICE) UK case definition for acute COVID-19 infection [2]; and (2) observational studies, including cohort, case-control and cross-sectional studies. We excluded studies that (1) were conducted before December 2019; (2) were case reports, opinions, commentaries and interventional studies; (3) included unconfirmed COVID-19 or other SARS-like illnesses; (4) involved animals; and (5) did not meet the inclusion criteria. Pre-print articles were considered acceptable if they met the inclusion criteria.

We anticipated studies reporting persistent symptom prevalence would involve participants reporting a gastrointestinal symptom during acute infection and its persistence beyond four weeks from diagnosis or beyond the resolution of other COVID-19 symptoms. While symptom persistence beyond twelve weeks may warrant a diagnosis of post-COVID syndrome, this diagnostic term was not used earlier in the pandemic and would not detect participants with post-viral symptoms lasting for between four and twelve weeks. Due to the nature of the review topic, we imposed no restrictions on participant age or comorbidity, and studies were not required to include a control or comparator group for inclusion. Two reviewers (MJH and NMV) independently screened the citations, and any discrepancies were resolved by consulting a third reviewer (DH).

### 2.3. Data Extraction

The following data were extracted into a spreadsheet for manual review.Study details: publication date, journal, authors, year, location, study design and setting (i.e., community or hospital) and funding source.Population characteristics including age, number of cases and controls, case definition, illness severity, vaccination status, SARS-CoV-2 variant and diagnostic criteria.Acute COVID-19 symptoms that relate to the gastrointestinal system.Point prevalence of persistent gastrointestinal symptoms after acute COVID-19, and the timepoint and method for which these symptoms were reported. Post-COVID symptoms reported would be persistent in nature, i.e., participants with a short-lived, unrelated episode of acute gastroenteritis at follow-up would not be captured.Incidence of new gastrointestinal illness presenting after recovery from acute COVID-19.

### 2.4. Risk of Bias and Quality Assessment

Studies eligible for inclusion were quality assessed using the Joanna Briggs Institute (JBI) Critical Appraisal Tools for observational studies [26]. Quality and risk of bias were assessed by two reviewers independently (MJH and NMV). The JBI Critical Appraisal Tools assess the methodological quality and risk of bias of each study, appraising aspects such as exposure and outcome measurement, controlling for confounding variables, loss of participants to follow-up and appropriateness of statistical analysis. The strength of methodological quality was graded by the percentage of positive answers, with studies scoring ≥ 70% considered high quality, studies scoring 50–69% considered moderate quality and studies scoring ≤ 49% considered low quality [27,28].

### 2.5. Data Analysis

We conducted a narrative synthesis of the studies to summarise the characteristics of each study, method of diagnosis and gastrointestinal-specific outcome measurements. We then conducted a descriptive analysis of all studies and reported the point prevalence for each symptom and the time at which this was captured. Where possible, persistent symptoms were grouped into one of four categories: diarrhoea (including loose stools and liquid stools); nausea and vomiting (including feeling and being sick); taste and smell disorders (including ageusia, dysgeusia, anosmia and altered taste and smell); and abdominal pain (including stomach ache/pain). We estimated the pooled prevalence for each symptom category across all studies by calculating the weighted average of the number of symptomatic cases divided by the total cohort size. We calculated all pooled prevalence estimates under the Freeman–Tukey double arcsine transformation and reported the 95% confidence intervals obtained under a random-effects model. Funnel plots were used to help identify heterogeneity and bias.

Change in point prevalence was visualised against time from acute COVID-19 resolution. Where participants reported more than one symptom per category (i.e., nausea, vomiting or haematemesis), the highest reported prevalence was used in the analysis. Overall pooled prevalence for any gastrointestinal complaint was only calculated from studies which reported this specifically and not calculated from the raw data to avoid double counting of participants reporting multiple symptoms. We also calculated the median symptom prevalence across all studies and for studies deemed to be at low risk of methodological bias, including interquartile ranges and median follow-up times.

For studies reporting the prevalence of gastrointestinal symptoms in both COVID-19 cases and controls, we calculated the odds ratio of having persistent gastrointestinal symptoms in a COVID-19 cohort vs. healthy controls for each symptom in each study. We calculated the I^2^ statistic to assess heterogeneity between studies and conducted both common and random effects meta-analyses to estimate overall odds ratios and 95% confidence intervals using the ‘metabin’ function in R package ‘meta’, version 6.0-0 in R version 4.1.0 [29].

## 3. Results

The initial search after duplicate removal identified 2549 potentially relevant studies; of these, 45 were eligible for inclusion in the review (Figure 1). Each reported the point prevalence of persistent gastrointestinal symptoms at specific timepoints, up to a maximum of 18 months from SARS-CoV-2 infection. The studies were conducted from early 2020 to mid-2023 in twenty-eight different countries across Europe, the Americas, Africa and Asia (Table 2 and Table 3). Forty-one studies were sourced from peer-review journals [8,30,31,32,33,34,35,36,37,38,39,40,41,42,43,44,45,46,47,48,49,50,51,52,53,54,55,56,57,58,59,60,61,62,63,64,65,66,67,68,69]. Four studies were pre-print and were awaiting peer review at the time of writing [70,71,72,73].

### 3.1. Study Characteristics and Outcome Measurement

Most studies (*n* = 35) followed up participants from COVID-19 diagnosis or acute illness to recovery (Table 2 and Table 3) and assessed for post-viral symptoms thereafter. Four studies followed up patients for a different clinical reason, such as nutritional status and endoscopy, and assessed post-COVID symptoms as a secondary outcome. The remaining six studies were cross-sectional and reported the point prevalence of a persistent symptom in participants who had historically tested positive for SARS-CoV-2 [30,31,32,55,63,70]. Three of these studies had a maximum possible duration between diagnosis and inclusion in the study of 16 months, although, for most participants, this was less than six months [30,31,32]. One cross-sectional study did not specify the study period or time to follow up [70]. Prevalence studies were generally conducted from early 2020 through to 2023, whereas studies reporting incidence were conducted from early 2022 to 2023.

Studies varied in how they defined COVID-19 cases. Twenty-eight studies included only participants with a positive reverse transcription polymerase chain reaction test (RT-PCR) for SARS-CoV-2 [32,41,42,43,44,45,46,47,48,49,50,51,52,53,54,55,57,58,60,61,62,65,67,68,69,71,72,73]. Seven studies included participants who tested positive via an unspecified diagnostic method [30,33,56,59,64,66,70]. Two database studies included participants with an ICD-10 code for confirmed COVID-19 in their electronic health record (EHR) [8,60]. Three studies included patients with COVID-19 pneumonia diagnosed through a computerised tomography (CT) scan in addition to those diagnosed via RT-PCR [34,35,36]. Two included those diagnosed with clinical symptoms in addition to those diagnosed via RT-PCR and CT scan, but this formed a small percentage of the overall cohort (4% and 16%) [31,37]. One study included participants with COVID-19 illness diagnosed based on clinical history and examination by a physician, in addition to those with laboratory-confirmed SARS-CoV-2 infection [38]. Two studies included participants with positive anti-SARS-CoV-2 antibodies on serological testing, with one also including participants diagnosed via RT-PCR [39,40]. One study included participants reporting either a positive RT-PCR or SARS-CoV-2 rapid antigen test. 

Only one study reported SARS-CoV-2 vaccination status; this is reported as a descriptive statistic, however, and is not a covariate used in the analysis [56]. It is likely the vast majority of participants included in other studies were not vaccinated due to the timing of studies in relation to vaccine rollout globally. The studies also varied in their clinical settings and ascertainment of gastrointestinal symptoms. Fourteen studies were conducted on patients who were hospitalised with COVID-19, discharged and followed up in the community [33,34,35,41,42,43,44,45,57,60,62,63,67,71], twenty-one studies involved both hospitalised and community-managed cases [8,30,32,36,37,40,46,47,48,49,50,55,59,61,64,65,66,68,69,70,72], seven studies involved community cases that did not require hospital admission [38,39,51,52,53,56,58] and three studies did not specify the setting of recruitment [31,54,73].

Most studies (*n* = 27) measured gastrointestinal outcomes using a survey or questionnaire administered via telephone or electronically [30,31,32,36,37,38,39,40,45,46,47,49,50,51,52,53,54,55,56,57,58,59,61,63,65,70,71]. Seven studies used EHRs to identify gastrointestinal endpoints based on clinical codes [8,60,66,67,69,72,73]. Eight studies involved an in-person clinical assessment by a healthcare professional [33,34,35,41,43,44,64,68]. One study involved both a questionnaire and clinical assessment with a healthcare professional [48]. Two studies measured outcomes through both clinical assessment and endoscopy [42,62].

#### Symptom Prevalence

Overall, the average prevalence of persistent post-COVID gastrointestinal symptoms of any nature and duration was 10.8% when weighted by cohort size (*n* = 111,198 across seven studies). For two studies reporting the total prevalence of persistent gastrointestinal symptoms of any nature and duration in healthy controls, the weighted average prevalence was 4.9% (*n* = 106,710). The median prevalence for all symptoms across all studies reporting prevalence data was 5.2% (IQR 9.3; 1.2–10.5, n = 36 studies). For seven studies reporting prevalence and deemed to be at low risk of methodological bias, prevalence estimates ranged from 0.2% to 24.1%, with a median follow-up time of 18 weeks.

Figure 2 shows the prevalence of persistent gastrointestinal symptoms over time in adults. We distinguished between studies that report point prevalence at a specific time point and studies that reported prevalence with minimum symptom duration (i.e., diarrhoea at 12 weeks vs. diarrhoea for at least 12 weeks). For studies reporting symptom prevalence at multiple timepoints, we selected the latest timepoint to avoid double counting of participants. We calculated pooled symptom prevalence for studies reporting diarrhoea, nausea and vomiting, taste and smell disorders and abdominal pain, persisting for between one and 18 months (Table 4). 

Four studies reported the prevalence of diarrhoea, nausea and vomiting, taste and smell disorders and abdominal pain in both a SARS-CoV-2 exposed cohort and an unexposed control group. Funnel plots showed asymmetry, primarily driven by two larger studies in children, which show a protective effect for SARS-CoV-2 exposure against persistent gastrointestinal symptoms [52,55]. Another large study in adults that suggested SARS-CoV-2 to be protective against persistent nausea and diarrhoea, however, found taste and smell disorders to be strongly associated with SARS-CoV-2 exposure. Individual funnel plots for each symptom category are provided as a supplement (Appendix A). Each funnel plot shows a pooled estimate calculated under a random effects model, which we do not report due to significant inter-study heterogeneity. However, we report individual odds ratios with 95% confidence intervals for each study (Figure 3).

### 3.2. Studies with the Highest Quality Score

Six studies in adults scored very highly for methodological quality and were deemed to be at low risk of bias (Appendix A). The majority of studies scored poorly due to the absence of a comparator group. Noviello et al. compared persistent gastrointestinal symptoms between a cohort five months after acute COVID-19 and a cohort of healthy controls using the Structured Assessment of Gastrointestinal Symptoms (SAGIS) questionnaire, a validated tool used to identify the presence of a broad range of gastrointestinal illnesses [49,74]. There was a higher prevalence of diarrhoea in the COVID-19 cohort at five months than in healthy controls (21.2% vs. 9.6%, *p* = 0.05). While the incidence of IBS was similar in both cohorts (adjusted risk ratio 1.07 [95% CI: 0.72–1.60]), the prevalence of diarrhoea was higher in the COVID-19 cohort (adjusted risk ratio 1.88 [95% CI: 0.99–3.54]) [49].

Liptak et al. followed up with patients with moderate to severe COVID-19 and patients with mild COVID-19 seven months after infection and compared them with test-negative controls who visited a hospital emergency department. They reported at least one gastrointestinal symptom in 19% of the moderate to severe group and 7.3% of the mild group, while only 3.0% of the controls had any gastrointestinal symptom (*p* ≤ 0.01). Diarrhoea and abdominal pain were more prevalent in the COVID-19 cohorts vs. test-negative controls (*p* < 0.05 for diarrhoea and *p* < 0.001 for abdominal pain) [47].

Zhang et al. investigated the relationship between gastrointestinal symptoms in acute COVID-19 and the subsequent risk of functional gastrointestinal disorders (FGIDs). They reported a higher incidence of FGIDs at six months in SARS-CoV-2 positive vs. SARS-CoV-2 negative-serology cohorts (8.9% vs. 3.12%, *p* = 0.025). In multivariate analysis, the development of FGIDs was shown to be associated with gastrointestinal symptoms at COVID-19 onset [68].

Robineau et al. assessed for post-COVID syndrome symptoms using a questionnaire in 25,910 participants after performing home-dried blood spot testing for anti-SARS-CoV-2 antibodies. The presence of nausea, diarrhoea and constipation was weakly associated with SARS-CoV-2 seronegativity (adjusted odds ratios: 0.68 [95% CI: 0.16–1.95] for nausea; 0.61 [95% CI: 0.26–1.27] for diarrhoea; and 0.78 [95% CI: 0.42–1.33] for constipation). Abdominal pain lasting more than two months was significantly associated with negative SARS-CoV-2 serology (adjusted OR 0.42 [95% CI: 0.21–0.74], *p* = 0.006). Persistent anosmia or dysgeusia was strongly associated with seropositivity (adjusted OR 8.89 [95% CI: 6.03–13.28], *p* < 0.0001) [40].

Chang et al. conducted a database cohort study to investigate associations between SARS-CoV-2 and subsequent risk for inflammatory bowel disease (IBD) and coeliac disease while adjusting for healthcare-seeking behaviour by using a historical control cohort and propensity-score matching. They identified a hazard ratio for incident inflammatory bowel disease and coeliac disease diagnoses of 1.78 (95%CI: 1.72–1.84) and 2.68 (95%CI: 2.51–2.85), respectively.

### 3.3. Studies Conducted in Specific Patient Groups

Four studies were conducted in patient cohorts. One study reported the prevalence of post-COVID symptoms in a cohort of 222 IBD patients and found that 42.3% of ulcerative colitis (UC) and 45.9% of Crohn’s disease (CD) patients had symptoms lasting for more than twelve weeks. The most common persistent symptoms were abdominal pain (~11% in CD and ~4% in UC), diarrhoea (~8% in CD and ~3% in UC) and nausea (~7% in CD and ~3% in UC). Vomiting was less common (~0–1% in both groups). Symptoms persistence beyond 12 weeks was associated with discontinuation of immunosuppressive therapy in UC patients and initial hospitalisation in CD patients [46].

Dagher et al. reported gastrointestinal symptoms in 36.9% of a cohort of cancer patients after SARS-CoV-2 infection, with a median symptom duration of seven months. Post-COVID symptoms of any nature were not found to be associated with cancer type, leucopenia, age or need for hospital admission during acute COVID-19, however [59].

Belkacemi et al. investigated persistent post-COVID symptoms in 1217 unvaccinated end-stage renal disease patients undergoing renal replacement therapy (RRT) and reported the point prevalence of diarrhoea and taste and smell disorders at six months as 6% and 2.3%, respectively (*n* = 216). The risk of having persisting clinical symptoms at six months was higher in patients who were hospitalised with moderate to severe disease (1.64 times) and those requiring intensive care treatment (5.03 times). Older age and longer duration of dialysis were also found to increase the risk of persistent symptoms [37]. Another study on RRT and kidney transplant patients conducted in Thailand reported the prevalence of anorexia and abdominal pain at 90.9% and 62.5%, respectively, after three months. This study also found older age to be associated with gastrointestinal manifestations of post-COVID syndrome [57].

### 3.4. Children

Five studies included children under 18 years old. Borch et al. reported that, in children aged 6–17 years, diarrhoea and nausea were significantly less prevalent in the SARS-CoV-2 exposed group than in the control group (OR: 0.37, 95% CI: 0.18–0.73) one month after acute COVID-19 recovery [52]. Penner et al. found that 6.5% and 2.6% of children who had been hospitalised for paediatric multisystem inflammatory syndrome (*n* = 46) experienced abdominal pain and diarrhoea, respectively, at six months. This study also reported raised calprotectin, a biomarker found in faeces indicating gut inflammation, in 31% of children at six weeks and 7% at six months [43].

At 24 weeks post-COVID, Sedik et al. reported abdominal pain in 2% of children (*n* = 105) and that most children recovered quickly without significant sequelae [65]. A cross-sectional study conducted by Adler et al., however, identified a statistically significant higher prevalence of abdominal pain in children with a history of COVID-19 compared with those without (9.5% vs. 3.8%, *p* < 0.001) [55]. Both studies found persistent symptoms to be more prevalent in children aged over 11 years. Ahn et al. conducted a case-control study in younger children (median age 3, IQR 1.0–9.0) and identified abdominal pain as the most commonly reported post-COVID symptom persisting for more than two months [56].

### 3.5. Incidence of Gastrointestinal Illness

Eleven studies reported the incidence of new gastrointestinal illness following SARS-CoV-2 infection. This included various illnesses encompassing functional disorders, motility disorders, hepatic and biliary disorders, autoimmune-mediated illness and infection. 

Stepan et al. found that children who visited the emergency department with abdominal pain and tested positive for SARS-CoV-2 either six months before or three months after had a higher incidence of irritable bowel syndrome (IBS) than those who tested negative (91.3% vs. 54.5%, *p* = 0.044) [44].

Ghoshal et al. reported the incidence of new post-infection IBS, uninvestigated dyspepsia (UD) and IBS-UD overlap to be 5.3%, 2.1% and 1.8%, respectively (*n* = 280), in those who tested positive for SARS-CoV-2 six months prior. Kaplan–Meier analysis showed a higher probability of developing such illnesses following SARS-CoV-2 exposure vs. in healthy controls [50]. Austhof et al. reported the incidence of post-infection IBS to be 20.4% in a smaller study of 49 participants who completed a survey at six months, with half of these meeting the Rome IV diagnostic criteria for IBS [54]. In another study, multivariate logistic regression identified the presence of nausea and diarrhoea during acute SARS-CoV-2 infection as predictors for the development of IBS (OR 4.00 95% CI: 1.01–15.84 and OR 5.64 95% CI: 1.21–26.31, respectively) [67].

Xu et al. ascertained the incident diagnoses of gastro-oesophageal reflux disease, peptic ulcer disease, acute pancreatitis, functional dyspepsia, acute gastritis, IBS and cholangitis in a cohort 30 days post-acute COVID-19 and a contemporary and pre-pandemic control cohort. The hazard ratio for a composite of all incident diagnoses was 1.37 (95% CI: 1.33–1.14), with cholangitis exhibiting the highest hazard of 2.02 (1.55–2.63). The COVID-19 cohort also displayed a higher hazard for abnormal coagulation studies and deranged liver function tests (HR 1.59 95% CI: 1.52–1.65 and HR 1.30 95% CI: 1.28–1.32) [66]. In a larger study, Ma et al. also reported a higher hazard for FGIDs, peptic ulcer disease, gastro-oesophageal reflux disease, gallbladder disease, severe liver disease, non-alcoholic fatty liver disease and pancreatic disease [72].

Two studies reported the incidence of autoimmune-mediated gastrointestinal illness following previous SARS-CoV-2 exposure. One study reported a hazard of 1.40 (95%CI 1.02–1.90) for IBD. Another reported a hazard of 1.78 (95%CI: 1.72–1.84) for IBD and 2.68 (95%CI: 2.51–2.85) for coeliac disease, the latter adjusted for healthcare-seeking behaviour [69,72].

## 4. Discussion

In this systematic review of 45 studies and 2,224,790 patients, we identified a relatively low (median = 5.2%) and variable (IQR 9.3; 1.2–10.5, n = 36 studies) prevalence of persistent gastrointestinal symptoms following SARS-CoV-2 infection. Prevalence estimates for persistent symptoms of any duration ranged from 0.2% to 24.1% for seven studies judged to be at low risk of bias, with a median follow-up time of 18 weeks. A higher rate of incident gastrointestinal illness, including functional disorders, motility disorders, autoimmune-mediated illness and hepatic and biliary disorders, were also observed after SARS-CoV-2 infection.

We found extensive inter-study heterogeneity, as expected when synthesising data from multiple observational studies across a variety of settings in changing circumstances. The studies with the highest methodological quality indicated that substantial numbers of individuals may be susceptible to persistent post-COVID gastrointestinal symptoms. We identified a higher pooled prevalence of diarrhoea, abdominal pain, nausea and vomiting in adults previously exposed to SARS-CoV-2 compared with controls. However, we do not report a reliable pooled effect estimate due to inter-study heterogeneity. Given the rapid spread of SARS-CoV-2 globally and recent UK estimates of post-COVID syndrome prevalence of 3.3%, a substantial proportion of those with chronic gastrointestinal complaints in the general population may be attributable to SARS-CoV-2 [3,75].

The prevalence of post-COVID syndrome may vary depending on the SARS-CoV-2 variant and the healthcare-seeking behaviour of the population. COVID-19 illness severity and hospitalisation rates are known to differ between the original wild-type SARS-CoV-2 and later variants [76,77]. As such, SARS-CoV-2 variant epochs may account for much of the variation in symptom prevalence over time and place that we found in this review. One study in our review reported the SARS-CoV-2 vaccination rate; however, this was not used as a covariate in any analysis [56]. Healthcare-seeking behaviour was also known to change throughout the COVID-19 pandemic; healthcare avoidance may result in an underestimation of the true frequency of gastrointestinal events in a population, particularly for studies requiring an in-person clinical review [78].

Of the studies reporting the incidence of gastrointestinal illness post-COVID, one found that, in children attending the emergency department with functional abdominal pain, those with historical SARS-CoV-2 infection were significantly more likely to be diagnosed with IBS than abdominal migraine compared with pre-pandemic controls [44]. This finding is in keeping with the hypothesis that SARS-CoV-2 infection may trigger post-infection functional gastrointestinal symptoms, whereas the aetiology of abdominal migraine is more likely to be non-infectious [79]. Similarly, four studies reported a higher incidence of IBS between three and six months after COVID-19; however, one study in our review did not find any significant difference.

In contrast, a cohort study of children assessing self-reported symptoms suggested that previous SARS-CoV-2 infection protects against persistent diarrhoea and nausea one month after subsequent COVID-19 diagnosis. Although younger child age has been shown to be protective against COVID-19 mortality, this study did not consider that SARS-CoV-2-positive children and their families were likely to be isolated from other causes of infectious diarrhoea during the study period [52,80].

Two studies in our review reported an association between SARS-CoV-2 infection and the development of autoimmune-mediated gastrointestinal illness. This may partly be explained by the molecular mimicry hypothesis, suggesting that SARS-CoV-2 dysregulates the hosts’ humoral immune system as a key factor in inducing autoimmunity in predisposed individuals [81]. Given the lag time between SARS-CoV-2 infection and the development and diagnosis of illnesses such as IBD and coeliac disease, it is possible that the increased incidence of such illnesses may not be evident for several years.

It was not possible to accurately estimate the impact of prior SARS-CoV-2 exposure on the future risk of gastrointestinal infections based on the data presented in our review. Although limited, current research suggests that rates of *Clostridium difficile* infection are no different between SARS-CoV-2-exposed and -unexposed patients [82]. One study in our review reported no significant association between previous SARS-CoV-2 infection and hospitalisation for gastrointestinal infections in older adults. Ascertaining cases of gastrointestinal infection in the general population is complicated, however, by the low frequency of testing and self-limiting symptoms, with relatively few cases presenting to secondary healthcare. It is also likely that non-pharmaceutical interventions (NPIs) intended to reduce COVID-19 transmission have also significantly reduced the transmission and case rates of enteric infections [83]. Therefore, if there is any excess risk and burden of enteric disease associated with SARS-CoV-2 infection, this would not be detected in these studies and may warrant further longitudinal follow-up and investigation. 

Most studies did not control for antibiotic exposure during acute SARS-CoV-2 infection, despite up to three-quarters of COVID-19 patients receiving antibiotics early in the pandemic [84]. One study included in our review found that individuals treated with antibiotics during acute infection were more likely to have persistent diarrhoea than those who did not; another identified antibiotic exposure as the strongest predictor for post-COVID gastrointestinal sequelae [47,49]. Interestingly, Ghoshal et al. reported that all patients with persistent dysphagia received either oral or intravenous antibiotics during acute COVID-19 [50]. None of these studies, however, reported the incidence of bacterial co-infection in patients who received antibiotics.

Although not included in our initial extraction criteria, post hoc analysis identified only two studies reporting the incidence of low mood among patients with persistent post-COVID gastrointestinal symptoms, or vice-versa. Both studies did not find any significant associations between persistent gastrointestinal symptoms and symptoms of low mood, anxiety and sleep disturbance [49,68]. This is surprising considering that a higher prevalence of mental health conditions has been reported among those with functional gastrointestinal disorders during COVID-19 lockdowns and emerging evidence of mechanisms linking depression and gut health [85,86].

## 5. Limitations

Our review aimed to estimate the prevalence of persistent post-COVID gastrointestinal symptoms, but we faced several methodological challenges. Post-COVID syndrome is a dynamic condition that relapses and remits over time, so point prevalence estimates may not capture its true burden [87]. Half of the studies included were based on hospital cohorts of patients suffering more severe disease, which may overestimate symptom prevalence due to ascertainment bias. Our review supports the observations that post-COVID syndrome is more common in patients with a history of severe COVID-19 [37,47]. Hospitalised patients would generally be more likely to suffer comorbidities, including presentations with gastrointestinal symptoms [88]. Finally, six cross-sectional studies included in our review may be at a higher risk of recall bias, given the nature of recalling details of COVID-19 infection and providing a clinical history of gastrointestinal complaints retrospectively [30,31,55,63,70,71].

## 6. Conclusions

Our review found a generally low prevalence of persistent gastrointestinal symptoms up to eighteen months after COVID-19 recovery. However, most studies lacked comparator groups, so we could not determine whether this differs from background rates of gastrointestinal illness in the general population. We did identify—albeit from limited data—that individuals previously exposed to SARS-CoV-2 may be more likely to develop gastrointestinal illnesses, including IBS, dyspepsia, hepatic and biliary disease and autoimmune-mediated illnesses, than the general population. Significant heterogeneity between studies overall prevented us from providing reliable pooled estimates of long-lasting gastrointestinal consequences of SARS-CoV-2 infection. Ideally, COVID-19 studies would have included prospective observation of SARS-CoV-2-infected participants for the development of gastrointestinal complaints, accounting for vaccination, antimicrobial use, variant epochs and public health interventions. Given the established links between gut dysbiosis and a wide range of viral infections, new studies should also monitor the excess risk of enteric infections after the removal of COVID-related NPIs, and future pandemic preparedness would do well to include proactive surveillance of gastrointestinal infections.

## Figures and Tables

**Figure 1 viruses-15-01625-f001:**
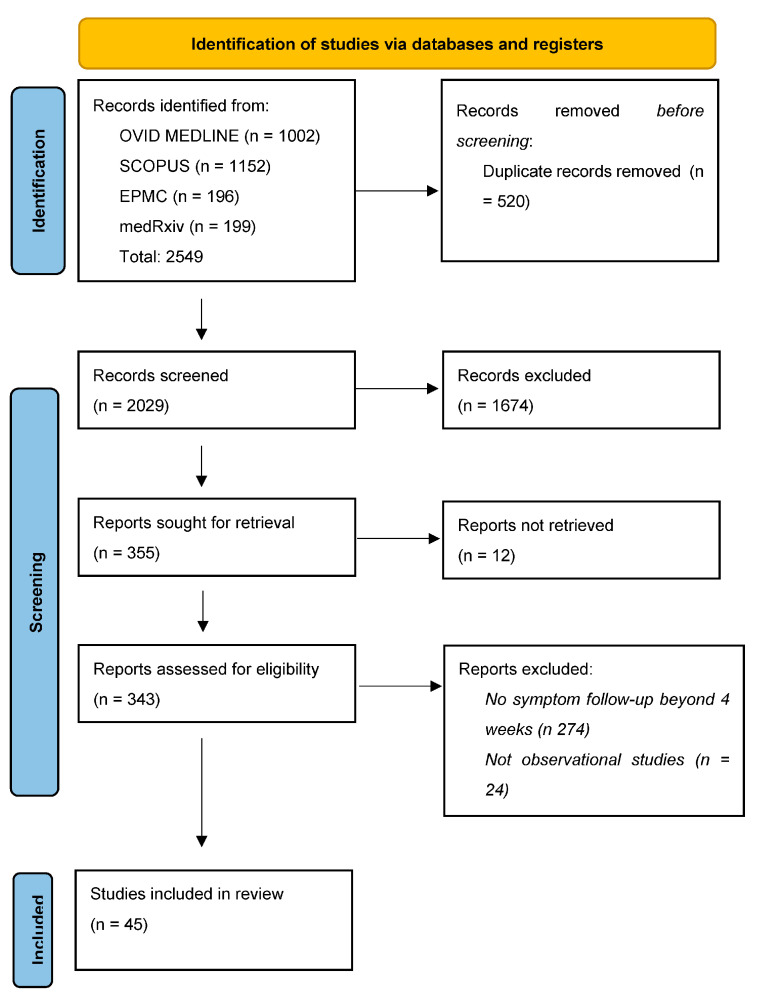
PRISMA flowchart detailing search results.

**Figure 2 viruses-15-01625-f002:**
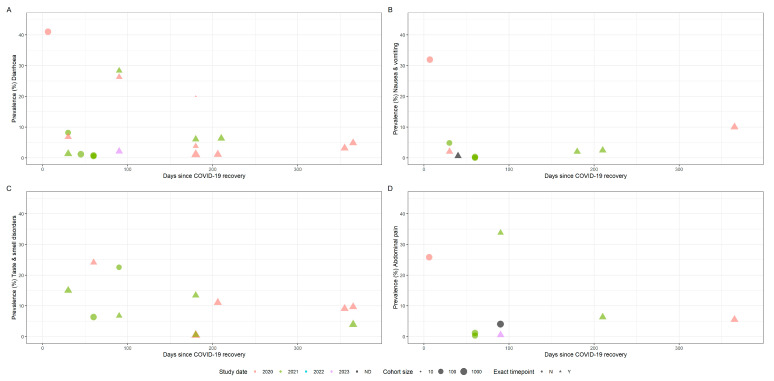
Time plots showing symptom prevalence against time since acute SARS-CoV-2 infection in adults (**A**: diarrhoea, **B**: nausea & vomiting, **C**: taste & smell disorders, **D**: abdominal pain). Size of points indicates cohort size. Triangles indicate point prevalence at that exact timepoint, whereas circles indicate studies reporting the lower bound of symptom duration. Colour indicates the study end date by year, whereas black indicates no study end date was specified.

**Figure 3 viruses-15-01625-f003:**
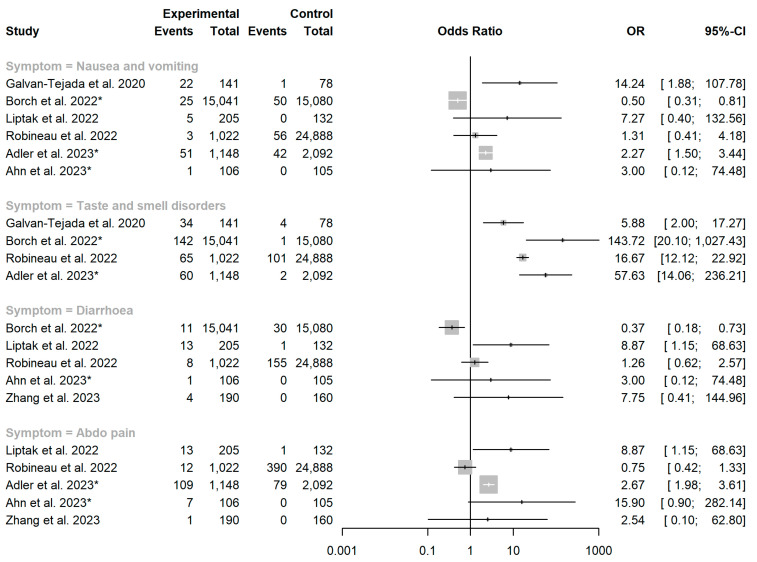
Forest plot showing the association between SARS-CoV-2 exposure and persistent gastrointestinal manifestations. OR = odds ratio. Asterisk (*) denotes studies conducted in children aged <18 years. Studies included: [40,47,51,52,55,56,68].

**Table 1 viruses-15-01625-t001:** Overview of search strategy, including limits.

Search Strategy	
1	“COVID-19” [Mesh] OR “SARS CoV-2” OR “coronavirus disease 2019”
2	“Gastrointestinal Disease” [Mesh] OR “gastrointestinal symptoms” OR “abdominal pain” OR “nausea” OR “vomiting” OR “diarrh?ea” OR “Diarrhea, Infantile” [Mesh] OR “constipation” OR “malnutrition” OR “gastritis” OR “gastroesophageal reflux” OR “pancreatitis” OR “colitis” OR “Deglutition Disorders” [Mesh] OR “?esophagitis” OR “transaminitis” OR “cholestasis” OR “cholestatic liver injury” OR “appendicitis” OR Gastrointestinal Haemorrhage” [Mesh] OR “upper gastrointestinal bleed” OR “lower gastrointestinal bleed”
3	“Longitudinal Studies” [Mesh] OR “longitudinal” OR “Cross-Sectional Studies” [Mesh] OR “cross-sectional” OR “Cohort Studies” OR “cohort” OR “Case-Control Studies” [Mesh] OR “case control” OR “Observational Studies” [Mesh] OR “observational”
4	1 AND 2 AND 3
5	Limit to humans and English language
6	4 AND 5

**Table 2 viruses-15-01625-t002:** Descriptive summary of studies reporting the point prevalence of persistent gastrointestinal symptoms after acute COVID-19.

Author	Publication Date	Country	Study Date	Study Design	Clinical Setting	Participant Age	Number of Cases	Case Definition	Acute Symptom Prevalence	Time Since Acute COVID-19	Persistent Symptom Prevalence
Islam, M. et al. [30]	February 2021	Bangladesh	One month (September to October 2020)	Cross-sectional without a comparator group	Community, some previously hospitalised	18–81 years, mean 34.7 (SD = 13.9)	1002	≥18 years old, having tested positive for SARS-CoV-2 and a willingness to complete the survey	Diarrhoea 27.3% Lack of appetite 57.4%	Not specified	Diarrhoea 6.9% Lack of appetite 11.4%
da Costa e Silva et al. [58]	January 2023	Brazil	July and August 2020	Cohort without a comparator group	Community	Mean 38.4 years	147	Healthcare and safety workers reporting one or more acute COVID-19 symptoms.	Nausea 26.5% Diarrhoea 40.1%	1 month	Nausea 2% Diarrhoea 6.8%
Liang, L. et al. [41]	December 2020	China	Three months (date not specified)	Prospective cohort without a comparator group	Hospitalised with community follow-up	24–76 years, mean 41.3 (SD = 13.8)	76	≥18 years old with lab-confirmed COVID-19, without a history of lung resection or neurological/psychiatric illness	Not reported	90 days	Diarrhoea 26.3%
Xie, XP. et al. [42]	September 2021	China	March toOctober 2020	Prospective cohort without a comparator group	Hospitalised with community follow-up	27–73 years	10	Lab-confirmed COVID-19 patients with GI symptoms	Diarrhoea 72.3% Nausea and vomiting 18.2% Anorexia 18.2%	6 months (diarrhoea) 3 months (abdominal symptoms)	Diarrhoea 20% Abdominal symptoms (abdominal pain, diarrhoea, constipation and others) 50%
Zhang et al. [68]	June 2023	China	July 2022 to February 2023	Prospective controlled cohort	Hospital and community	Mean 44.5 years	190	COVID-19 patients recruited from a dedicated COVID care centre in China.	Diarrhoea 8.9% Abdominal pain 4.2% Constipation 3.7% Dyspepsia 5.3% Overlap 4.7%	3 months	Diarrhoea 2.1% Abdominal pain 0.5% Constipation 2.1% Dyspepsia 2.1% Overlap 3.2%
Attauabi, M. et al. [46]	November 2021	Denmark	January 2020 to April 2021	Prospective cohort without a comparator group	Hospital and community	30–61 years	222 at follow-up	IBD patients with lab-confirmed COVID-19	Not reported	At least 12 weeks	Ageusia 22.5%
Borch, L. et al. [52]	January 2022	Denmark	24 March to 9 May 2021	Controlled cohort	Community	0–17 years	15041	Children 0–17 years old with lab-confirmed COVID-19	Not reported	More than 4 weeks	Nausea 0.2% Loss of taste 0.8% Loss of smell 0.9%
Vaillant, MF. et al. [33]	July 2021	France	May to July 2020	Prospective cohort without acomparator group	Hospital, with community follow-up	22–97 years	403 in total	Adult inpatients hospitalised with lab-confirmed COVID-19 who returned home after hospitalisation. Participants with persistent symptoms were all following an enriched or altered diet	Not reported	1 month	Nausea/vomiting 4.0% Anorexia/early satiety/long satiation 7.9% Anosmia/ageusia or dysgeusia/change in taste 8.7%
Faycal, A. et al. [39]	November 2021	France	10 March to 18 May 2020	Prospective cohort without a comparator group	Community	Median 41.6 years (IQR 30–51.5)	175 (28 to 60-day follow-up)	Symptomatic adult outpatients (>18 years), with lab confirmed COVID-19 or positive anti-SARS-CoV-2 antibodies	Not reported	30 days, 60 days for ageusia	GI symptoms 6.9% Ageusia 53.6%
Gerard, M. et al. [34]	November 2021	France	1 March to 29 April 2020	Prospective cohort without a comparator group	Hospitalised with community follow-up	Mean age 59.8 years	288 (53 to 6 months)	≥18 years old, lab and/or computerised tomography (CT) confirmed COVID-19 and discharged from hospital	Diarrhoea 9.4%	180 days	Diarrhoea 3.8%
Belkacemi, M. et al. [37]	March 2022	France	March to December 2020	Prospective cohort without a comparator group	Community and hospital	Not specified	1217	All dialysis patients reported lab- or CT-confirmed COVID-19 or suspicious clinical symptoms	Not reported	6 months	Diarrhoea 1.1% Persistent anosmia or dysgeusia 0.4%
Robineau, O. et al. [40]	April 2022	France	April 2020 to Janurary 2021	Population-based controlled cohort	Community	Range 33.5–61.0 years	1022	18 to 69 years old, positive for SARS-CoV-2 antibodies	Not reported	More than 2 months	Nausea 0.3% Diarrhoea 0.8% Constipation 1.6% Abdominal pain 1.2%
Augustin, M. et al. [48]	July 2021	Germany	6 April to 2 December 2020	Prospective cohort without a comparator group	Primarily community, with 2.9% hospitalised	31–54 years, mean 43	353 (958 at the acute stage)	≥18 years old, with lab-confirmed COVID-19	Diarrhoea 19.0%, Ageusia 59.1%	206 days	Diarrhoea 1.1% Ageusia 11.0%%
Noviello, D. et al. [49]	June 2021	Italy	February to August 2020	Prospective controlled cohort	Community and hospital	18–60 years	164	18 and 60 years with lab-confirmed COVID-19 and without a previous diagnosis of IBS, IBD or coeliac disease	Nausea 25% Diarrhoea 52% Abdominal pain 20% Sickness 10% Weight loss 50%	5 months	Symptom prevalence not reported, reported as adjusted SAGIS score difference *
Comelli, A. et al. [45]	March 2022	Italy	February 2020 to May 2021	Prospective cohort without a comparator group	Hospital with community follow-up	Mean 59.4 years	456	Adults with lab-confirmed COVID-19 were admitted to eight hospitals in North and Central Italy, excluding patients <18 years and/or pregnant	Not reported	12 months	Smell disorder 3.9% Taste disorder 2.9% Severe GI problems 0.2% Decreased appetite 7.5% Altered GI function (altered bowel habit and bloating) 32.7%
Damanti, S. et al. [35]	July 2022	Italy	24 August– to 6 June 2021	Prospective cohort without a comparator group	Community, previously hospitalised	>65 years	176	Age >65 years, previously hospitalised for COVID-19 pneumonia and discharged alive	Not reported	6 months	Dysgeusia 0.6% Anosmia 0.6%
Fatima, G. et al. [71]	Pre-print July 2021	India	Not specified	Cohort, without a comparator group	Community, previously hospitalised	17–88 years, mean 56	160	≥18 years old, with lab-confirmed COVID-19	Not reported	40 days	Nausea and vomiting 0.6% Loss of appetite 6.25%
Rao, G. et al. [70]	Pre-print July 2021	India	Not defined	Cross-sectional without a comparator group	Community and hospital	Not reported	2038	Not specified	Not reported	1–3 months	Abdominal pain 4.0% Digestive issues 10.3%
Adler et al. [55]	February 2023	Israel	December 2021 to January 2022	Cross-sectional controlled	Community, 0.4% hospitalised	Range 5–18 years	1148	Children aged 5–18 years with a positive PCR test for SARS-CoV-2 one to six months prior to data collection.	Not reported	Not specified	Abdominal pain 9.5% Reduced taste 5.2% Nausea 4.4%
Sedik et al. [65]	June 2023	Iraq	July to September 2021	Prospective uncontrolled cohort	Hospital and community	Median 6.3 years	105	Children aged <16 years who visited a paediatric teaching hospital in Iraq with confirmed COVID-19	Gastrointestinal symptoms 72.4% Abdominal pain 42.9% Nausea 29.5% Vomiting 40% Diarrhoea 45.7% Decreased bowel motion 2.9%	More than 24 weeks	Abdominal pain 2% (no other symptoms reported)
Imoto et al. [63]	December 2022	Japan	1 January to 31 December 2020	Cross-sectional	Hospitalised with community follow up	Median 60 years	285	All patients diagnosed with SARS-CoV-2 infection or hospitalised with COVID-19 at each hospital in Osaka	Dysguesua 39% Anosmia 38% Lack of appetite 54% Diarrhoea 19%	11.7 months	Dysguesia 9% Anosmia 9% Lack of appetite 8% Diarrhoea 3%
Fischer et al. [61]	August 2022	Luxembourg	1 May to 8 November 2020	Cohort, without a comparator group	Hospital and community	Mean 40.2 years	289	Patients with a history of a positive SARS-CoV-2 RT-PCR test performed at one of five laboratories in Luxembourg	Not reported	12 months	Stomach burn 7.6% Abdominal pain 5.6% Feeling sick 10% Diarrhoea 4.8% Loss of smell 9.7%
Fernandez-Plata, R. et al. [53]	August 2022	Mexico	6 April 2021 to 14 December 2021	Prospective cohort, without a comparator group	Community	Range 29–45 years	149	Workers at the National Institute of Respiratory Diseases with lab-confirmed COVID-19	Not reported	6 months	Diarrhoea 6.0% Nausea 2.0% Dry mouth 7.4% Mouth ulcers 3.4% Bile alteration 13.4% Weight changes 10.7% Dysgeusia/ageusia 10.7% Anosmia 13.4%
Galvan-Tejada, C. et al. [51]	December 2020	Mexico	25 July to20 September 2020	Case-control	Not specified	Mean 39 years	141	Lab confirmed COVID-19, and at least 14 days since positive test and enrolment	Not reported	Up to 60 days	Anosmia or dysgeusia 24.1% Nausea, vomiting or diarrhoea 15.6%
Qamar, M. et al. [32]	February 2022	Pakistan	Nov 2020 toApril 2021	Cross-sectional without a comparator group	Community, some previously hospitalised	Range 18–35 years	331	≥18 years old with lab-confirmed ≥1 month ago	Diarrhoea 26.3% Loss of appetite 36.3% Nausea 17.8%	More than one month	Diarrhoea 8.2% Loss of appetite 13.0% Nausea/vomiting 4.8%
Khodeir, M. et al. [31]	December 2021	Saudi Arabia	September to October 2020	Cross-sectional without a comparator group	Not reported	Range 10–84 years	979	Recovered COVID-19 patients, with lab confirmed COVID-19 or clinical symptoms	Not reported	6–9 days	Diarrhoea 41.4% Nausea 32.0% Lack of appetite 46.5% Abdominal pain 25.9%
Liptak, P. et al. [47]	July 2022	Slovakia	February to October 2021	Prospective controlled cohort	Hospital and community	Range 32–68 years	205	Adult patients recruited from an outpatient COVID-19 testing centre, >18 years old and with lab-confirmed COVID-19	Diarrhoea 24.9% Abdominal pain 10.7% Bloating 5.4% Nausea 14.1% Heartburn 2.4%	7 months	Diarrhoea 6.3% Abdominal pain 6.3% Bloating 4.9% Nausea 2.4% Heartburn 3.4% Vomiting 1.0%
Ahn et al. [56]	May 2023	South Korea	May to July 2022	Case-control study	Outpatient clinic	Mean 3 years	106	Children older than six months visiting an outpatient clinic from 1 May to 31 July with a previous diagnosis of COVID-19.	Not reported	At least 12 weeks	Vomiting 0.9% Diarrhoea 0.6% Abdominal pain 6.6%
Fernandez-de-Las-Penas et al. [60]	May 2023	Spain	10 March to 31 May 2020	Cohort without a comparator group	Hospitalised with community follow up	Mean 61 years	1266	Hospitalised COVID-19 survivors whose ICD-10 diagnosis of SARS-CoV-2 was confirmed by RT-PCR during the first wave of the pandemic at five hospitals in Madrid	Diarrhoea 8.3% Anosmia 8.5% Aguesia 5.2% Vomiting 3%	18 months	Anosmia 0.64% Persistent gastrointestinal symptoms 2.4%
Chancharoenthana et al. [57]	May 2023	Thailand	January 2022 to 31 July 2022	Prospective cohort without a comparator group	Hospitalised	Mean 52 ±11 years	577	Dialysis-dependent patients and kidney transplant patients under the care of a renal referral tertiary care centre	Nausea or vomiting 29.6% Diarrhoea 19.2% Anosmia 12.8%	At least 3 months	Anorexia 90.9% Abdominal pain 62.5% Aguesia 64%
Karaarslan, F. et al. [36]	May 2021	Turkey	18 November 2020 to 20 January 2021	Prospective cohort without a comparator group	Community following hospital discharge	Mean 53 years	300	Age 18–70, discharged from hospital following lab- or CT-confirmed COVID-19, not requiring ITU admission	Loss of appetite 71.7% Diarrhoea 21.3% Loss of taste 53%	30 days	Loss of appetite 10.3% Diarrhoea 1.4% Loss of taste 15%
Penner, J. et al. [43]	July 2021	UK	4 April to 1 September 2020	Retrospective cohort without a comparator group	Hospitalised, with community follow-up	Range 0–18 years Median age: 10·2 (8·8–13·3)	46	Patients aged ≤18 years, fulfilling the UK Royal College of Paediatrics and Child Health (RCPCH) diagnostic criteria for PIMS-TS following lab-confirmed COVID-19	Abdominal pain, diarrhoea, vomiting or abnormal abdominal imaging 98%	6 months	Abdominal pain 6.5% Diarrhoea 2.6%
Austhof, E. et al. [54]	July 2022	USA	May 2020 to October 2021	Prospective cohort without a comparator group	Not reported	Mean 42.7 years	1449	>18 years with lab-confirmed COVID-19, recruited from the Arizona CoVHORT database	Not reported	>45 days	Acid reflux/heartburn 0.6% Anorexia 0.1% Early satiety 0.2% Feeling of not emptying bowels 0.2% Diarrhoea 1.2% Constipation 0.6% Other GI symptoms (not otherwise specified) 1.7%
Dagher et al. [59]	February 2023	USA	March 2020 to May 2021	Cohort without a comparator group	Community and hospital	Median 57 years	312	Patients with cancer receiving care at the University of Texas MD Anderson Cancer Center who were also diagnosed with COVID-19	Not reported	More than 1 month	Gastrointestinal symptoms 36.9% Abnormal smell or taste 28.5%
Taquet, M. et al. [8]	September 2021	USA	20 January to 16 December 2020	Retrospective controlled (influenza group) database cohort	Hospital and community	Mean 46.3 years	106,578	Clinical diagnosis of COVID-19 (ICD-10 code U07.1)	Not reported	3–6 months	Abdominal symptoms 10.69%
Wu, Q. et al. [38]	July 2022	USA	10 March 2020 to31 March 2021	Prospective cohort without a comparator group	Community	Mean 46 years	74	Individuals with lab-confirmed COVID-19, or COVID-19 diagnosed by a healthcare professional, from the Understanding America Study COVID-19 National Sample	Anosmia 43.2% Diarrhoea 47.3% Abdominal discomfort 37.8% Vomiting 14.9%	12 weeks	Anosmia 6.8% Diarrhoea 28.4% Abdominal discomfort 33.8% Diarrhoea 5.4%
Karuna et al. [64]	June 2023	USA, Peru, Malawi, South Africa, Zambia and Zimbabwe	May 2020 to March 2021	Prospective cohort without a comparator group	Hospital and community	Mean 45.1 years	578	Aged 18 years and older from the USA, Peru, Malawi, South Africa, Zambia and Zimbabwe with a history of symptomatic SARS-CoV-2 infection	Any gastrointestinal symptoms 68.7% Abdominal pain 17.1% Anorexia 46.7% Diarrhoea 39.6% Nausea/vomiting 27.9%	More than 60 days	Any gastrointestinal symptom 1% Abdominal pain 0.4% Anorexia 0.5% Diarrhoea 0.5% Nausea/vomiting 0%

SD: standard deviation; lab-confirmed COVID-19: positive reverse transcription polymerase chain reaction test for SARS-CoV-2; IBS: irritable bowel syndrome; IBD: inflammatory bowel disease; * SAGIS questionnaire score difference of +0.16 for abdominal pain/discomfort, +0.13 for diarrhoea/incontinence and +0.13 for gastroesophageal reflux disease/regurgitation, with higher scores in the previously SARS-CoV-2-exposed cohort.

**Table 3 viruses-15-01625-t003:** Descriptive summary of studies reporting the incidence of new gastrointestinal illness after acute SARS-CoV-2 infection.

Author	Publication Date	Country	Study Date	Study Design	Clinical Setting	Participant Age (Years)	Number of Cases	Case Definition	Acute Symptom Prevalence	Follow-Up Time	Incidence Rate of New Illness
Ghoshal, U.C. et al. [50]	November 2021	Bangladesh and India	April to August 2020	Prospective controlled cohort	Hospital and community	Median: 35.9	280	Lab-confirmed hospitalised and outpatient COVID-19 cases, excluding patients with prior history of FGIDs, abdominal surgery, psychiatric illness, IBD and gastrointestinal cancer	Nausea 18.9% Vomiting 10.4% Diarrhoea 20.7% Ageusia 35.4% Abdominal pain 11.1%	6 months	Uninvestigated dyspepsia 2.1% IBS 5.4% IBS-UD overlap 1.8%
Stepan, M. D. et al. [44]	March 2022	Romania	1 February to 1 August 2021	Retrospective controlled cohort	Community following presentation at hospital	Range 4–6	23	Preschool children presenting to ED with chronic abdominal pain, with a history of lab-confirmed COVID-19 3–6 months prior	Not reported	3–6 months	IBS 91.3% Abdominal migraine 8.7%
Penner, J. et al. [43]	July 2021	UK	4 April to 1 September 2020	Retrospective cohort without a comparator group	Hospital, with community follow-up	Range 0–18 Median age: 10·2 (8·8–13·3)	46	Patients aged ≤18 years, fulfilling the UK Royal College of Paediatrics and Child Health (RCPCH) diagnostic criteria for PIMS-TS following lab-confirmed COVID-19	Abdominal pain, diarrhoea, vomiting or abnormal abdominal imaging 98%	6 months	New diarrhoeal illness 2.2% New nausea and vomiting 2.2%
Austhof, E. et al. [54]	July 2022	USA	May 2020 to Oct 2021	Prospective cohort without a comparator group	Not reported	Mean: 42.7	49	>18 years with lab confirmed COVID-19, recruited from the Arizona CoVHORT database	Not reported	6 months	New post-infection IBS 20.4%
Golla et al. [62]	February 2023	India	April 2021 to January 2022	Prospective controlled cohort	Hospitalised with community follow up	Mean: 38	320	Patients admitted to a COVID-19 hospital in India recruited post-discharge	Gastrointestinal complaints 15.6%	6 months	FGID/DGBI 6.6%
Xu et al. [66]	March 2023	USA	1 March 2020 to 15 January 2022	Retrospective controlled cohort	Hospital and community	Mean 61.75 years	154,068	US Veterans enrolled in the Veterans Health Administration electronic healthcare database with a SARS-CoV-2 positive test and who survived the first 30 days	Not reported	Between 1 and 12 months (median 5 months)	Gastro-oesophageal reflux disease HR 1.35 (1.31–1.39) Peptic ulcer disease HR 1.62 (1.46–1.79) Acute pancreatitis HR 1.46 (1.23–1.75) Functional dyspepsia HR 1.36 (1.22–1.51) Acute gastritis HR 1.47 (1.25–1.86) IBS HR 1.54 (1.28–1.86) Cholangitis HR 2.02 (1.55–2.63)
Yamamoto et al. [67]	April 2023	Japan	January 2020 to October 2021	Retrospective cohort without a comparator group	Hospitalised with community follow up	Range 34–64 years	571	Patients aged >18 with a positive RT-PCR for SARS-CoV-2 admitted to hospital for COVID-19 oxygen requirement	Nausea 6.1% Vomiting 3% Diarrhoea 45% Constipation 43.1% Abdominal pain 0.9% Abdominal distention 1.6%	1 to 12 months (median 5 months)	IBS 2.1%
Zhang et al. [68]	June 2023	China	July 2022 to February 2023	Prospective controlled cohort	Hospital and community	Mean 44.5 years	190	COVID-19 patients recruited from a dedicated COVID care centre in China	Diarrhoea 8.9%, Abdominal pain 4.2% Constipation 3.7% Dyspepsia 5.3% Overlap 4.7%	6 months	IBS 1.1%, Functional dyspepsia 1.6%
Chang et al. [69]	January 2023	Multiple (not specified)	1 January to 31 December 2020	Retrospective controlled cohort	Hospital and community	Mean 45.1 years	887,455	Patients ≥ 18 years old who had at least two healthcare visits and had received PCR tests during the study period registered in the TriNetX database	Not reported	6 months	Coeliac disease aHR 2.68 (2.52–2.85) IBD aHR 1.78 (1.72–1.84)
Andersson et al. [73]	Pre-print April 2023	Denmark	1 January 2021 to 10 December 2022	Prospecticve controlled cohort	Not specified	Mean age 66.6 years	930,071	Database cohort representative of the Danish population, aged 50 years or older and no prior record of SARS-CoV-2 infection	Not reported	After 1 month	Hospitalisation for gastrointestinal infection IRR 1.28 (0.78–2.09)
Ma et al. [72]	Pre-print April 2023	UK	Up to 30 November 2020	Population-based controlled cohort	Hospital and community	Range 37 to 73 years	112,311	Participants recruited from the UK biobank database with a record of a positive SARS-CoV-2 test	Not reported	Median 254 days	FGID HR 1.95 (1.62–2.35) Peptic ulcer disease HR 1.27 (1.04–1.56) Gastro-oesophageal reflux disease HR 1.46 (1.34–1.58) IBD HR 1.40 (1.02–1.90) Gallbladder disease HR 1.28 (1.13–1.46) Severe liver disease HR 1.46 (1.12–1.90) Non-alcoholic fatty liver disease HR 1.33 (1.15–1.55) Pancreatic disease HR 1.43 (1.17–1.74)

IBD: inflammatory bowel disease; IBS: irritable bowel syndrome; UD: uninvestigated dyspepsia; IBS-UD: irritable bowel syndrome–uninvestigated dyspepsia overlap; FGIDs: functional gastrointestinal disorders; aHR: adjusted hazard ratio (with 95% confidence intervals in braces); HR: hazard ratio (with 95% confidence intervals in braces); IRR: incidence rate ratio (with 95% confidence intervals in braces); ED: emergency department.

**Table 4 viruses-15-01625-t004:** Average pooled prevalence (with 95% confidence interval) of persistent symptoms for any duration between 1 month to 18 months.

Symptom	Total Sample Size	Weighted Prevalence (95% CI)	Unweighted Prevalence
Diarrhoea	25,274 across 24 studies	2.7% (0.014; 0.052)	7.2%
Nausea and vomiting	20,408 across 14 studies	1.5% (0.0055; 0.0424)	5%
Taste and smell disorders	20,916 from 17 studies	9.6% (0.0490; 0.1816)	16.8%
Abdominal pain	7357 from 13 studies	5.7% (0.0230; 0.1339)	12.7%

95% CI: 95% confidence interval.

## Data Availability

All data produced in the present work are contained in the manuscript.

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
