# Peer review of "A Systematic Review of the Prevalence of Persistent Gastrointestinal Symptoms and Incidence of New Gastrointestinal Illness after Acute SARS-CoV-2 Infection"

_viruses, 2023, doi:10.3390/v15081625_

Round 1

Reviewer 1 Report

The study aimed to systematically review and synthesise the global literature on the prevalence of long-term gastrointestinal symptoms following acute SARS-CoV-2 infection, and on the incidence of gastrointestinal disease following acute SARS-CoV-2 infection. The study identified 28 different studies from around the world, and was able to estimate prevalence of overall and specific sub-groups of long-term gastrointestinal symptoms. They also identified 4 studies reporting on incidence. This was a rigorous and well-conducted systematic review on an important health topic.

General comments:

1.      The headline results presented at the start of the discussion section - median = 6.25%,  range 1.2–11.4% prevalence… 0.2% to 24.1% for six studies judged to be at a low risk of bias – were not clearly reported in the results section and it wasn’t clear how these were calculated. Please make this clear and make sure all results are reported in the results section

2.      It would have been good to see pooled estimates and confidence intervals calculated for prevalence of overall and sub-groups of GI symptoms – consider using a method such as Freeman-Tukey to do this

3.      While not conducting a meta-analysis of OR was adequately justified, I think the OR should still be presented, e.g. in a forest plot, without a pooled estimate

4.      No evaluation of publication bias was presented – this should be included, e.g. using a funnel plot

Specific comments:  

5.      Abstract: The Embase database is mentioned, but this is not mentioned in the main paper – remove if it was not used

6.      Abstract: Please provide confidence intervals for pooled prevalences

7.      Abstract: “We also identified the presence of functional gastrointestinal disorders in histori-cally SARS-CoV-2 exposed individuals.” This is a rather vague summary – can any more concrete results be given?

8.      Introduction, page 2, paragraph 2: please mention the countries and dates of previous studies mentioned

9.      Table 1: Exact searches used in each database should be presented in supplementary material

10.   Figure 1: duplicate articles should be recorded and removed at the first stage, not at the final stage

11.   Results, p7: It would be helpful to summarise the dates when studies were conducted

12.   Figure 2: It would be helpful to indicate on the plot what each panel shows, either with a subheading or on the y-axis

13.   Discussion, p27, para 1: “low (median = 6.25%) and variable (IQR = 10.2%; range 1.2–11.4%) prevalence” – give the number of studies.

14.   Discussion, p27, para 3: “we faced several methodological challenges” – I think these are better described as limitations

The English language was generally good, with occasional small errors in grammar

Reviewer 2 Report

Hawkings et al reported the prevalence and incidence of symptoms after acute COVID-19 infections. I appreciate authors used systematic review approach to summarize the published literatures. However, I have some comments to improve this study.

1. In the background, I cannot see the knowledge gaps on your research gaps, especially in many papers on this topic published before.

2. In the methods, please use Newcastle Ottawa Scales as the ROB assessment. In addition, updated search is necessary.

3. Please address how to deal the overlapping population among the included studies.

4. Please provide the subgroup analyses by the study period of the included studies, and these findings may be also good to see the change of your results over time.

5. No language limitation is suggested. If you find the language other than English, you should use Google translate.

6. In the results, I cannot believe no records identified from Google scholars. Also, I would like to see the forest plot of the meta-analysis, instead of using table.

7. The judgment of ROB from included studies should be presented in the supplementary files.

8. Please justify how to determine the incidence data from case-control study. For my knowledge, case-control study is not appropriate for the calculation of incidence.

None.

Round 2

Reviewer 2 Report

Thanks for your clarifications and revisions. I consider this manuscript is now acceptable.